# Angiotensin 1–7 Stimulates Proliferation of Lung Bronchoalveolar Progenitors—Implications for SARS-CoV-2 Infection

**DOI:** 10.3390/cells11132102

**Published:** 2022-07-02

**Authors:** Andrzej K. Ciechanowicz, Wen Xin Lay, Jefte Prado Paulino, Erika Suchocki, Susanne Leszczak, Christian Leszczak, Magdalena Kucia

**Affiliations:** 1Laboratory of Regenerative Medicine, Center for Preclinical Research and Technology, Medical University of Warsaw, 02-097 Warsaw, Poland; s079331@student.wum.edu.pl (W.X.L.); s075754@student.wum.edu.pl (J.P.P.); s075769@student.wum.edu.pl (E.S.); s075711@student.wum.edu.pl (S.L.); s075775@student.wum.edu.pl (C.L.); magdalena.kucia@wum.edu.pl (M.K.); 2Stem Cell Institute at James Graham Brown Cancer Center, University of Louisville, Louisville, KY 40202, USA

**Keywords:** lung epithelial cells, bronchoalveolar stem cells, alveolar type 2 cells, COVID-19, RAAS, ACE2, Nlrp3 inflammasome, angiotensin 1–7

## Abstract

SARS-CoV-2 infection leads to severe lung damage due to pneumonia and, in more severe cases, leads to acute respiratory distress syndrome, or ARDS. This affects the viability of bronchoalveolar cells. An important role in the pathogenesis of these complications is the hyperactivation of the renin-angiotensin-aldosterone (RAA) pathway and induction of cytokine storm that occurs in an Nlrp3 inflammasome-dependent manner. To shed more light on the susceptibility of lung tissue to SARS-CoV-2 infection, we evaluated murine bronchioalveolar stem cells (BASC), alveolar type II cells (AT2), and 3D-derived organoids expression of mRNA encoding genes involved in virus entry into cells, components of RAA, and genes that comprise elements of the Nlrp3 inflammasome pathway. We noticed that all these genes are expressed by lung alveolar stem cells and organoids-derived from these cells. Interestingly, all these cells express a high level of ACE2 that, on the one hand, serves as an entry receptor for SARS-CoV-2 and, on the other, converts angiotensin II into its physiological antagonist, angiotensin 1–7 (Ang 1–7), which has been reported to have a protective role in lung damage. To shed more light on the role of Ang 1–7 on lung tissue, we exposed lung-derived BASC and AT2 cells to this mediator of RAA and noticed that it increases the proliferation of these cells. Based on this, Ang 1–7 could be employed to alleviate the damage to lung alveolar stem/progenitor cells during SARS-CoV-2 infection.

## 1. Introduction

The SARS-coronavirus 2 (SARS-CoV-2) damages the lungs, blood vessels, heart, kidneys, brain and intestines, and increases inflammatory monocytes and neutrophils and causes an abrupt decrease in lymphocytes [1,2]. The disease starts initially with pulmonary symptoms, together with impairment of blood oxygenation that ultimately, with the progression of infection, leads to major dysfunction of the vascular epithelium. In fact, impairment of the alveolar-capillary barrier and increased endothelial permeability are major factors in the pathogenesis of acute respiratory distress syndrome (ARDS) [1,2].

SARS-CoV-2 enters human cells by binding to angiotensin-converting enzyme-2 (ACE2). Once attached to ACE2, the viral spike protein is primed by the host serine protease TMPRSS2 [3,4], which ultimately allows the fusion of viral and cellular membranes. CD147, an extracellular matrix metalloproteinase inducer, and sialic acid receptors have also been demonstrated as possible binding proteins for SARS-CoV-2. The overexpression of human ACE2 enhanced disease severity in a mouse model of SARS-CoV-2 infection and injecting SARS-CoV-2 spike into mice worsened lung injury [4]. Interestingly, this damage is attenuated by blocking the renin-angiotensin pathway and depends on ACE2 expression [1].

Recent advances have improved our understanding of the renin-angiotensin system (RAS), with the recognition that angiotensin (Ang)—(1–7) is a biologically active product of the RAS cascade. The RAS consists of a series of enzymatic reactions that result in the generation of angiotensin II (Ang II). In the first step, renin cleaves hepatic peptide angiotensinogen to produce Ang I in the blood. Subsequently, Ang I is hydrolyzed by the angiotensin-converting enzyme (ACE), producing the octapeptide Ang II. Ang II acts on Ang II type 1 and type 2 receptors (AT_1_R and AT_2_R). Ang II is the central effector of the RAS and exerts biological activities, including proliferative and pro-fibrotic actions, vasoconstriction, and consequent blood pressure elevation. In addition to this direct action, chronically elevated Ang II stimulates several pathophysiological mechanisms, including the generation of oxidative stress, stimulation of the nervous system, alteration in renal hemodynamics, and activation of the immune system [1,3,4,5,6]. 

Of importance to this report, ACE2, zinc metallopeptidase and a homologue of ACE are known to be responsible for producing Ang (1–7) by cleaving Ang II. Ang (1–7) binds and activates G protein-coupled Mas receptor, which is linked to G_aI_, to exert anti-inflammatory effects [7,8,9]. The ACE2/Ang (1–7)/Mas receptor pathway often serves to counter-regulate the bl pathway [9]. Therefore, Ang (1–7) seems to be critical in protecting against lung inflammation and fibrosis, since it inhibits alveolar cell apoptosis, attenuates endothelial cell activation, and limits the synthesis of pro-inflammatory and profibrotic cytokines [7,8,9]. In models of pulmonary inflammation, the administration of Ang (1–7) results in a decrease in inflammatory cell migration and overall improvement in pulmonary function [8,9].

The binding of SARS-CoV-2 to ACE2 downregulates its expression, leading to an increase in Ang II. ACE2 is expressed particularly abundantly in alveolar epithelial cells and vascular endothelial cells [10]. It has already been demonstrated that hyperactivation of AT_1_R by Ang II leads to redundant activation of Nlrp3 inflammasome and cell death by pyroptosis in lung epithelial cells, endothelium, and cardiomyocytes [10]. 

SARS-CoV-2 infection may lead to a “cytokine storm,” due to the activation of Nlrp3 inflammasome. This intracellular pattern recognition receptor, consisting of Nlrp3 scaffold, an adaptor apoptosis speck-like protein (ASC), and the effector procaspase-1, initiates the formation of inflammasome by interacting with ASC, which recruits and activates procaspase-1 to generate active caspase-1 and then converts the cytokine precursors pro-IL-1beta and pro-IL-18 into mature and biologically active IL-1 beta and IL-18, respectively. Once activated, Nlrp3 will trigger a series of inflammatory responses and pyroptotic cell death [10]. It is well known that the innate immune response and activation of the Nlrp3 inflammasome are important defense mechanisms during the first days of infection, until acquired immunity responds with the production of antibodies. However, on the other hand, hyperactivation of this intracellular protein complex may induce a cytokine storm, with detrimental effects, leading to multi-organ failure.

Herein, we report that murine bronchioalveolar stem cells (BASC), alveolar type II cells (AT2), and 3D-derived organoids express mRNA encoding genes involved in virus entry into cells, components of RAA, and genes that comprise elements of the Nlrp3 inflammasome pathway. We also demonstrate that ACE-2 product Ang (1–7) stimulates proliferation of bronchioalveolar stem cells (BASC), alveolar type II cells (AT2), and 3D-derived organoids, which supports that it could be employed to ameliorate damage to lung alveolar stem/progenitor cells during SARS-CoV-2 infection. 

## 2. Material and Methods

### 2.1. Mice

Mice were kept under constant temperature conditions (22 °C ± 2 °C) and controlled humidity (55% ± 10%) and lighting (12 h light and 12 h of darkness) at the Central Animal Laboratory of the Medical University of Warsaw. The experiment was approved by the Local Bioethics Commission, and all procedures on mice were carried out by researchers who had approval from the Medical University of Warsaw. 

Wild type (C57BL/6Clzd) mice were purchased from the Central Animal Laboratory of Medical University of Warsaw. The experiment was carried out on healthy male wild-type mice 6–8 weeks old with an average weight of 21 g (±1 g). After two weeks of acclimatization, the animals were divided in a randomized manner and assigned to control and experimental groups (*n* = 5).

### 2.2. Administration of BrdU and Angiotensin 1–7

Mice were injected intraperitoneally with BrdU (BD, #550891) in a volume of 90 µL daily for 15 days. In addition, from the fourth to the fourteenth day inclusive, the mice were injected subcutaneously with a solution of Angiotensin 1–7 (0.5 mg/kg bw) suspended in a volume of 100 µL of sterile PBS. In the following 2 h after BrdU administration, Angiotensin 1–7 was administered. On the last fifteenth day, mice were sacrificed 1 h after BrdU administration.

### 2.3. Bronchoalveolar Lavage Fluid Collection

Bronchoalveolar lavage fluid (BALF) was collected from mice after anesthesia with urethane. For each mouse, the lung was washed with 0.6 mL sterile PBS twice and pooled together. To collect cells, BALF was centrifuged for 5 min at 800× *g*, then 15 min at 1500× *g* to remove cell debris. BALF protein concentration was measured by Bradford assay.

### 2.4. Lung Harvest and Lung Single-Cell Suspension

For the analysis, 15 days after the start of BrdU injections, mice were anesthetized with ketamine/xylazine, followed by thoracotomy and right ventricular perfusion to remove blood cells from the alveolar space, as described previously [11]. The middle lung lobe was tied off. The remaining lung was inflated with 3 mL collagenase IV (LS004212, Worthington biomedical corporation, Lakewood, NJ, USA) in DMEM (Dulbecco’s modified Eagle’s medium) followed by 1% low melting agarose (AB00981, American Bio, Natick, MA, USA) and left for 1 min for cooling. The lung was then digested with collagenase IV for 30 min in a water bath at 36 °C, dissociated using gentle MACS tissue dissociator (Miltenyi Biotec, Auburn, CA, USA), and incubated with DNase (100 units/mL; Roche, Basel, Switzerland) for 15 min in a water bath at 36 °C. Cells were then filtered through 70- and 40-µm cell strainers. Cells were washed with DMEM and processed for flow cytometry analysis and cell sorting.

### 2.5. Isolation of Murine Alveolar Type II Cells and Bronchioalveolar Stem Cells by FACS and Flow Cytometry BrdU Analysis

AT2 and BASCs sorting was performed as previously described [12]. (The number of wild-type mice used for cell sorting *n* = 6). Briefly, single lung cells suspension was stained with the following anti-mouse antibodies (all antibodies were provided by BD, Franklin Lakes, NJ, USA): APC anti-CD45 (Clone: 30-F11), APC anti-CD31 (Clone: MEC 13.3), and PE-Cy7 anti-CD326 (EP-CAM) (Clone: G8.8) and PE anti-Ly-6A/E (Clone: E13-161.7). After staining, cells were washed once and re-suspended in PBS containing 2% FBS. AT2 (CD31 negative, CD45 negative, Sca-1 negative, Ep-CAM positive) and BASCs (CD31 negative, CD45 negative, Sca-1 positive, Ep-CAM positive) were sorted with the use of FACS MoFlo (Beckman Coulter, Brea, CA, USA), (Figure 1). In addition, after staining according to the same protocol as for cell isolation using FACS, BrdU staining and analysis using flow cytometry were performed.

### 2.6. Lung Organoid Model System

In order to evaluate whether AT2 cells and BASCs in the 3D model show expression of the studied genes, we established an organoid culture based on McQualter et al. [13] and Teisanu et al. [14], revised and modified by Barkauskas et al. [15]. Organoid cultures were conducted in 24-well Transwells with 0.4 µm pores filter inserts (Corning, Corning, NY, USA) in a 24-well tissue culture plate containing 410 μL of the 3D medium. A 3D medium was prepared with DMEM F12 (Dulbecco’s Modified Eagle’s Medium/F12) (88%, *v*/*v*) supplemented with FBS (10%, *v*/*v*), 1 M HEPES (0.1%, *v*/*v*), penicillin/streptomycin (0.5%, *v*/*v*), and insulin/transferrin/selenium (1%, *v*/*v*) for all cultures. FACS sorted AT2 cells (3 × 10^3^) and BASCs (1 × 10^3^) (per single 0.4 µm insert) were co-cultured with (1 × 10^5^) MLG cells (Mlg2908, ATCC CCL-206, Manassas, VI, USA) in Matrigel (Growth Factor Reduced (GFR) Basement Membrane Matrix, Phenol Red-free, *LDEV-free; #356239; Corning). Ice-cold matrigel was prediluted 1:1 with ice-cold 3D medium. Cultures were incubated at 37 °C in a humidified incubator (5% CO_2_), with medium replacement every other day for 21 days. After this time, organoids were fixed and paraffin embedded, then cut as 3 µm sections and H&E stain. Organoids then were visualized, and images were taken with 10×, and 20× objectives using a NIB-100 (Alltion, Wuzhou, China) light microscope and imaged using an ISH 1000 (TUSCEN) camera and Iscopture V3.0 software. Image analyses were performed using NIH ImageJ software. version bundled with 64-bit Java 1.8.0_172.

### 2.7. Detection of RAA System and Nlrp3 Inflammasome Genes in Lung Epithelial Cells and Lung 3D Organoids

In order evaluate Nlrp3 inflammasome and RAA system genes, we performed quantitative RT-PCR (*n* = 3) for the expression of ACE, ACE2, CD147, MAS, TOP, AT1, AT2, AT3, AGT, TMPRSS2, REN, Nlrp3, CASP1, IL-1 beta, IL-18, AIM2, GSDM, NR1D1, HMGB1, S100A9, NEP, CYT1 and CYT2 in isolated AT2 cells, BASCs and lung organoid lysate. The primer sequences that were used are listed in Appendix A. The expression of the genes listed above was quantified by real-time PCR using SYBR Green Super Mix (Bio-Rad, #1725124, Hercules, CA, USA). All PCRs were performed using the following conditions: pre-denaturation at 95 °C for 3 min, and 40 cycles of denaturation at 95 °C for 10 s, and annealing at 60 °C for 60 s. The expression of each gen was normalized to beta-2-microglobulin.

### 2.8. LC-MS Proteome Analysis

A normalized concentration of proteins from each sample (BAL fluid and lung tissue lysates) was precipitated with the use of ice cold (−20 °C) Acetonitrile (ACN, Merck, Readington Township, NJ, USA) in volume 1:4 ratio. After precipitation, samples were centrifuged (−9 °C, 30 min, 18,000× *g*), the supernatant removed, and excess of ACN was evaporated using a vacuum centrifuge (5 min, room temp.). The protein pellet was dissolved in 40 mM ammonium bicarbonate. Reduction and alkylation were carried out using 500 mM DTT (in final concentration 20 mM) and 1M IAA (in final concentration 40 mM). Proteins were in-solution digested for 16 h incubation with Trypsin Gold (Promega, Madison, WI, USA) in 37 °C. Digested samples were diluted with 0.1% formic acid (ThermoFisher, Waltham, MA, USA) and centrifuged (+2 °C, 30 min, 18,000× *g*) before nano-UHPLC separation.

LC-MS analysis was carried out with the use of nano-UHPLC (nanoElute, Bruker, Billerica, MA, USA) coupled by CaptiveSpray (Bruker) to an ESI-Q-TOF mass spectrometer (Compact, Bruker). The two-Column separation method was used, i.e., pre-column (300 µm × 5 mm, C18 PepMap 100, 5 µm, 100 Å, Thermo Scientific, Waltham, MA, USA) and Aurora separation column with CSI fitting (75 µm × 250 mm, C18 1.6 µm) in gradient 2% B to 35% B for 90 min with a 300 nL/min flow rate. The following mobile phases were used: A—0.1% formic acid in water; B—0.1% formic acid in ACN. 

Ionization of the samples was carried out at a gas flow of 3.0 L/min, a temperature of 150 °C, and voltage of the capillary 1600 V. The quadrupole energy was set to 5.0 eV, and collision chamber energy 7.0 eV with an ion transfer time of 90 µs. The ions were analyzed in the positive polarity mode in the range 150–2200 m/*z*, with an acquisition frequency of the 1 Hz spectrum, as well as with the autoMS/MS system.

The collected spectra were analyzed and calibrated (Na Formate) in DataAnalysis software (Bruker) and then, after extracting the compound list, identified in ProteinScape (Bruker) using the MASCOT server. Proteins were identified using the online SwissProt (www.uniprot.org accessed on 7 May 2022) and NCBI_prot databases (https://www.ncbi.nlm.nih.gov/ accessed on 7 May 2022), and their annotation and biological significance were identified using UniProt.org, Reactome.org, String.org, and KEGG.org.

### 2.9. Statistical Analysis 

Data were analyzed with Prism 6 (GraphPad Software, San Diego, CA, USA) as mean ± SD. Statistical analysis was performed using a two-tailed unpaired Student’s *t* test to compare the differences between two groups. *p* < 0.05 was considered to be statistically significant.

## 3. Results

### 3.1. Purified Alveolar Rype 2 cells, Bronchoalveolar Stem Cells, and Lung Organoids Express mRNA for SARS-CoV-2 Entry Proteins and RAAS Peptides

We employed qRT-PCR to study the expression of mRNA for SARS-CoV-2 entry proteins and RAAS peptides in murine primary FACS-sorted AT2 cells and BASCs and, subsequently, 3D lung organoids derived from AT2 and BASC cells (AT2 co-cultured with MLG cells, and BASCs co-cultured with MLG cells) (Figure 1). All qRT-PCR results were normalized to the expression of the tested genes in the mouse lung fibroblast cell line MLG (Mlg2908, ATCC CCL-206).

Figure 2 shows that we found very high expression of the SARS-CoV-2 entry receptor ACE2 in the purified primary AT2 cells compared to MLG reference cells, as well in primary BASCs. Moreover, ACE2 expression was detected in cells derived from organoids plated from AT2 and BASCs. At the same time, all studied cells expressed a very high-level host serine protease TMPRSS2 involved in virus entry and CD147, known as basigin or EMMPRIN, which facilitates SARS-CoV-2 infection. However, CD147 was expressed highly on BASCs and lower on AT2 cells (Figure 2). We also detected the expression of several genes involved in RAAS signaling, including the Mas receptor (MAS) that binds Ang(1–7), renin, an enzyme that processes the conversion of angiotensinogen into Ang I, angiotensin converting enzyme (ACE), neural endopeptidase (NEP), angiotensin II (AGT), angiotensin II type 1 receptor (AT1R), angiotensin II type 2 receptor (AT2R) and AT3R. The mRNA encoding these proteins was particularly highly expressed by BASC. 

### 3.2. Purified Alveolar Type 2 Cells, Bronchoalveolar Stem Cells and Lung Organoids Express mRNA for Components of the Nlrp3 Inflammasome Complex

We have previously proposed that SARS-CoV-2 hyperactivates the Nlrp3 inflammasome and by the induction of cytokine storms in SARS-CoV-2-infected patients, leads to multi-organ damage [16,17,18]. To address whether lung epithelial cells may activate inflammasome-related responses, we evaluated the first expression of mRNA for several inflammasomes on primary murine BASC and AT2 cells, as well as in 3D organoids plated from AT2 cells and bronchoalveolar stem cells (Figure 3). We noticed the high expression of Nlrp3 inflammasome, as well as nuclear receptor subfamily 1 group D member 1 (*NR1D1*), in lung epithelial BASC cells. There was also high expression of *AIM2* in lung epithelial BASC and AT2 cells. 

Our qRT-PCR data demonstrated also high expression of Nlrp3 inflammasome components in murine BASC and AT2 cells, as well as in 3D organoids-derived from these cells, including mRNA for caspase 1, IL-1beta and IL-18 (Figure 3). These cells also expressed gasdermin D (*GSDM*) involved in cell damage by pyroptosis and mRNA for two alarmines *HMGB1* and *S100A9*.

### 3.3. Ang (1–7) Enhances Proliferation of Murine Lung BASCs and AT2 Cells

After we noticed that murine lung BASC and AT2 cells express several RAA peptides and receptors (Figure 2), and as reported, the ACE2 product, Ang (1–7), has a protective role in lung damage, we asked whether it might directly increase in vivo proliferation of lung stem/progenitor cells. To address this question, male mice were exposed to daily subcutaneous injection for 11 days of Ang (1–7). In parallel with Ang (1–7), mice were administered BrdU intraperitoneally. The control group received vehicle instead of Ang (1–7), but were also administered BrdU. We found that 11-day administration of Ang (1–7) evaluated in this experiment directly stimulated proliferation of BASC and AT2 cells in vivo, as assessed by the percentage of cells that incorporated BrdU. Figure 4 shows an increase in BrdU accumulation by lung-residing BASC from 1.36 ± 0.93% to 3.30 ± 1.39%. At the same time, the number of BrdU-accumulating AT2 cells changed compared with the control from the 0.26 ± 0.18% to 0.74 ± 0.30% level. These results support that Ang (1–7) may directly stimulate lung stem/progenitor cells. 

### 3.4. Effect of In Vivo Administration of Ang (1–7) on the Lung Tissue Proteome 

To evaluate the protein profile of lung tissue changes after angiotensin (1–7) administration, we employed mass spectrometry and identified 1832 proteins, out of which 836 were further analyzed (Figure 5). The basis for the selection of proteins subjected to detailed proteomic analysis was their detection in experimental groups. Only proteins present in 51% of samples classified into one group were analyzed in further detail. Such an approach was taken to remove so-called individual differences between samples. Therefore, the proteomic profile was studied focusing on experimental group differences.

Among the proteins with statistically significant variable expression, 26 proteins were observed, the expression of which changed compared to the control group in the range from −7.36 fold change to +3.99 fold change (Figure 6A). In addition, there were 26 proteins that were silenced after administration of angiotensin 1–7, as well as 21 proteins that were only seen in the angiotensin (1–7) stimulated group (Figure 6F). 

### 3.5. Identified Proteins with Variable Expression Were Analyzed Using the UniProt, KEGG and Reactome Online Databases

Proteins involved in the induction of apoptosis and/or pyroptosis were observed among proteins with down-regulated or completely silenced expression after the administration of angiotensin (1–7). To this group was assigned caspase-6 (CASP6), and thioredoxin-dependent peroxide reductase (PRDX3), methylthioribulose-1-phosphate dehydratase (MTNB), reticulon-4 (RTN4), and lymphocyte-specific protein 1 (LSP1), which were completely silenced (Figure 5).

Administration of angiotensin (1–7) to mice also resulted in the expression of proteins that protect against apoptosis, which were found in the group of control mice, including apolipoprotein E (APOE), ADP-ribosylation factor 4 (ARF4), and syntaxin-binding protein 1 (STXB1).

After administration of Ang (1–7), we found an increase in the expression of proteins responsible for cell proliferation and differentiation, eukaryotic translation initiation factor 5A (IF5A2) protein, neuroplastin (NPTN), and copine-1 (CPNE1). Additionally, after Ang (1–7), we detected Ras-related protein Rab-5A (RAB5A) and Cadherin-13 (CAD13), which were not found in the control animals. 

Notably, after Ang (1–7), we found much lower concentration or complete silencing of proteins negatively regulating cell proliferation and differentiation, neuropilin-1 (NRP1), CD44 antigen (CD44), dystrophin (DMD), nucleoside diphosphate kinase A (NDKA) and dadherin-1 (CADH1) (Figure 5).

In addition, we performed a protein analysis using STRING, indicating the interactions and relationships of protein groups (Figure 6B). As a result of this analysis, we observed that Angiotensin (1–7) stimulation changed the expression of protein groups annotated to the interleukin-1 signaling pathway, cytoskeleton reorganization, and peptide metabolic process, which is consistent with the changes described above.

All proteins identified in whole lung tissue lysate were assigned to specific biological processes (using UniProt and Reactome) (Figure 6D). Among the identified proteins, the most numerous groups of proteins were those involved in metabolism and its regulation (150 proteins, of which 22 had statistically significantly altered expression), immune system (137 proteins, 5 with statistically significantly altered expression), metabolism of proteins (98 proteins, of which 13 had statistically altered expression), and responsible for signal Transduction (91 proteins, of which 11 had statistically significantly altered expression).

Of the proteins involved in the immune system signaling pathways (Figure 6E), the most (57 proteins) were annotated to neutrophil degranulation, which is part of the innate immune system. This group of signaling pathways was also represented by the Fc gamma receptor (FCGR)-dependent phagocytosis pathway with 14 proteins annotated, the C-type lectin receptors (CLRs) pathway with 11 proteins annotated, and the Fc epsilon receptor (FCERI) signaling pathway with 10 proteins annotated. Detailed UniProt analysis also revealed that proteins annotated to the signaling pathway, involving interleukins (45 proteins) and FLT3 signaling (15 proteins), which are pathways that are classified as “cytokine signaling in the immune system”. Furthermore, among the “adaptive immune system” category, 14 proteins were assigned to Class I MHC mediated antigen processing and presentation, and 9 proteins were assigned to the signaling via B Cell Receptor (BCR) pathway.

### 3.6. The Effect of the In Vivo Administration of Ang (1–7) on the Bronchoalveolar Lavage Proteome 

1028 proteins was identified in the collected BAL fluid, of which 482 proteins were further analyzed (Figure 5). Of the proteins analyzed in angiotensin (1–7)-stimulated mice, the expression of 42 proteins changed statistically significantly, 46 were silenced entirely, and 32 were expressed after Ang (1–7) administration (Figure 6F). The observed statistically significant changes in protein expression ranged from a −7.03-fold change to a 4.70 fold change (Figure 6A).

For BAL fluid after Ang (1–7) administration, we observed a reduction in the expression of proteins involved in programmed cell death, including advanced glycosylation end product-specific receptor (RAGE). In addition, cytochrome c oxidase subunit 2 (COX2), cathepsin L1 (CATL1), cell adhesion molecule 1 (CADM1), calpain-1 catalytic subunit (CAN1), cytosolic phospholipase A2 (PA24A) and non-specific lipid-transfer protein (NLTP) completely silenced in BAL fluid of mice that received Ang (1–7) (Figure 5). 

Ang (1–7) administration resulted in increased the expression of many proteins involved in the positive regulation of lung cell differentiation and proliferation, including adhesion G protein-coupled receptor F4 (AGFR4), unconventional myosin-Va (MYO5A), prelamin-A/C (LMNA), keratin, type II cytoskeletal 2 epidermal (K22E), phosphatidylethanolamine-binding protein 1 (PEBP1), myotrophin (MTPN), keratin, type II cytoskeletal 6A (K2C6A), peroxiredoxin-2 (PRDX2), purine nucleoside phosphorylase (PNPH), filamin-A (FLNA), keratin, type II cytoskeletal 8 (K2C8), cadherin-13 (CAD13) and transforming growth factor-beta-induced protein ig-h3 (BGH3) (Figure 5).

In addition, stimulation of mice with angiotensin (1–7) resulted in the expression of proteins involved in the positive regulation of proliferation and differentiation of lung epithelial cells that were not detected for BAL fluid of the control group, i.e., pro-epidermal growth factor (EGF), TRPM8 channel-associated factor 2 (TCAF2), 40S ribosomal protein S14 (RS14), 2’,3’-cyclic-nucleotide 3’-phosphodiesterase (CN37), interferon alpha/beta receptor 2 (INAR2), roquin-2 (RC3H2), DCC-interacting protein 13-beta (DP13B), mediator of RNA polymerase II transcription subunit 1 (MED1), dixin (DIXC1), keratin, type I cytoskeletal 16 (K1C16) and sialic acid-binding Ig-like lectin 10 (SIG10). 

In parallel, we detected a decrease in the expression of the advanced glycosylation end product-specific receptor (RAGE) protein, which is involved in the negative regulation of the cell proliferation process. 

Next, we analyzed the interdependence and grouping between proteins with variable expression in BAL Fluid (Figure 6C). There are three main groups of proteins with common biological functions, i.e., MAPK family signaling cascades, regulation of immune system process, and cytoskeleton organization. Stimulation of mice with angiotensin (1–7) influences, via these signaling pathways, biological processes, i.e., apoptosis, cell differentiation, and proliferation, which is in line with the results observed in the mRNA expression profile and cytometric analysis.

## 4. Discussion

The silent observation of our study is that murine lung stem/progenitor cells highly express mRNA for SARS-CoV-2 entry receptors, RAAS peptides, and Nlrp3 inflammasome. This indicates the involvement of lung tissues in regulating the activity of the RAAS system and immune responses, which explains the complexity of lung involvement in SARS-CoV-2-mediated infection. Importantly, we also noticed that Ang (1–7) directly stimulates in vivo proliferation of lung stem/progenitor cells. Several studies have demonstrated that treatment with Ang (1–7) activates anti-inflammatory responses, improves arterial oxygenation, reduces the expression of genes involved in collagen expression in the lungs, and decreases collagen deposition in the lung tissue [1,3,9]. Herein, we demonstrate for the first time that bronchioalveolar stem cells (BASC) and alveolar type II cells (AT2) are stimulated by Ang (1–7). 

BASCs are a lung resident stem cell population located at bronchioalveolar duct junctions that contribute to the maintenance of bronchiolar club cells and alveolar epithelial cells of the distal lung [19]. BASCs differentiate into alveolar-type (AT)-1 cells, AT2 cells, club cells, and ciliated cells. In response to terminal-bronchiole injury, the number of BASCs increases resulting tissue repair [19]. To date, the functions of BASCs have been mainly assessed in models of bleomycin-induced lung injury, hyperoxia, and influenza virus infection [19]. SARS-CoV-2 infection of BASCs can potentially lead to defects in lung regeneration capacity. It was demonstrated that ACE2 is co-expressed with elements of the kinin–kallikrein, renin–angiotensin and coagulation systems in alveolar cells [19]. A meta-analysis of a healthy human lung cell atlas with ~ 130,000 public single-cell transcriptomes demonstrated that crucial elements of the bradykinin, angiotensin, and coagulation systems are co-expressed with ACE2 in alveolar cells and associated with their differentiation dynamics [20]. Here, to determine whether lung BASCs can be infected by SARS-CoV-2, we analyzed the expression of SARS-CoV-2 entry factors in murine primary sorted lung epithelial stem cells, as well 3D organoids derived from both studied populations. We found that BASCs and AT2 stem cells express ACE2, TMPRSS2, and CD147, making them susceptible to SARS-2-CoV-2 infection, which may result in a decreased capacity for lung regeneration and delayed recovery from disease. 

In recent years, the Ang (1–7)/Mas receptor axis has gained interest as a possible strategy to treat SARS-CoV-2 infection [10]. In our experiments, Ang (1–7) administration led to an increase in the number of lung-residing BASC and AT2 cells. Experimental and clinical evidence indicate that the activation of Ang (1–7)/Mas receptor is an important mechanism to ameliorate the damaging effects triggered by overactivation in AngII/At1 receptor interactions [21]. Ang (1–7) increases eNOS expression via Akt-dependent pathways, resulting in endothelium-dependent vasodilation [22]. In addition, Ang (1–7) stimulates the proliferation of endothelial progenitor cells, regenerating the injured endothelial layer. Ang (1–7) also causes a great reduction in thrombus formation. 

In a well-structured study using a murine experimental model of asthma, Ang (1–7) activated mechanisms crucial for the resolution of the inflammatory process [23]. It was demonstrated that Ang (1–7) promoted the return of lung homeostasis via the decreased expression of NF-kB in eosinophils, reduction in GATA3, ERK1/2, and IκB-α expression in the lungs and decreased pulmonary remodeling and collagen deposition [23]. Our results showed that treatment with Ang (1–7) resulted in a decreased level or lack of proteins involved in the induction of apoptosis and pyroptosis in lung tissue, including caspase 6, PRDX3, LSP1, RTN4 and MTNB. Furthermore, mass spectrometry analysis revealed an increased number of proteins annotated to the group of positive regulators of cell proliferation and differentiation and a decreased number of proteins annotated to negative regulators of proliferation phenotype after Ang (1–7) treatment.

The ACE2/Ang 1–7)/MasR axis plays an important role in the biology of vasculogenic progenitor cells, hematopoietic stem cells, cardiomyocytes and dental stem cells [10,24,25]. Ang (1–7) improves pancreatic β-cell function and long-term Ang (1–7) administration in vivo affects insulin secretion from islets [26]. It was demonstrated that activation of MasR in CD34^+^ cells obtained from healthy individuals promoted vascular repair–relevant functions, such as migration, proliferation, and NO generation. Our bronchoalveolar lavage proteome results also indicate that Ang (1–7) positively modulates lung epithelial stem cell differentiation and proliferation.

Our results are highly relevant to the potential long-term effects of SARS-CoV-2 infection, as the virus may damage BASC and AT2 cells by (i) direct cell entry and subsequent cell lysis or (ii) induction of cell death by pyroptosis due to hyperactivation of the Nlrp3 inflammasome in response to S protein or to excessive Ang II stimulation by AT_1_R. An inflammasome is defined by its sensor protein, which oligomerizes to form a pro-caspase-1-activating platform in response to danger associated molecular patterns (DAMPs) or pathogen-associated molecular patterns (PAMPs). Nlrp3 inflammasome is, so far, the best-studied member of the inflammasome family, consisting of several members.

Nlrp3 is a tripartite protein that consists of an Nlrp3 protein, an apoptosis-associated speck-like protein containing a CARD (ASC), and pro-caspase 1 [27]. Upon activation, it becomes an aggregate composed of several Nlrp3 molecules (speck complexes), each containing Nlrp3 protein, ASC, and pro-caspase 1. Nlrp3-mediated activation of the caspase-1 promotes proteolytic cleavage and maturation of two pro-inflammatory cytokines, interleukin-1beta (IL-1beta) and interleukin 18 (IL-18), and cleavage of gasdermin-D, which forms cell membrane pores involved in the secretion of mature IL-1beta and IL-18. Altogether, this leads to the release of several alarmines or DAMPs that subsequently amplify an uncontrolled immune response [17]. This response involves secretion from other cells of several pro-inflammatory cytokines such as IL-6 or TNF-α and mediators, as well as the activation of the complement and coagulation cascades [28]. If uncontrolled, this process may end in a cytokine storm and fatal organ damage. We have recently demonstrated that SARS-CoV-2 spike protein induction of Nlrp3 inflammasome leads to damage to hematopoietic stem progenitor cells.

Recently, NLRP3 inflammasome has been demonstrated as an important contributor to various fibrotic diseases, following the detection of its constant activation [29] and elevated activation of the Nlrp3 in the lung tissue of both pulmonary fibrosis mice and IPF patients [10,29]. Specifically, NLRP3 was observed in epithelial cells in the fibrotic lung tissue [29], and its activation promoted myofibroblast differentiation of lung-resident mesenchymal stem cells. Our MS analysis revealed the potential involvement of Nlrp3 in lung epithelial cells, since we detected after Ang (1–7) stimulation changes in the expression of proteins annotated to interleukin-1 signaling pathway, which is a final executor of Nlrp3 activation.

In conclusion, our study has several limitations and further research is required to extend these findings. We confirmed the expression of RAAS components in lung epithelial BASCs, alveolar type 2 cells, as well as 3D organoids derived from these two primary murine populations. We also demonstrated the expression of key factors involved in Nlrp3 signaling in these cells. This shed more light on lung tissue and its involvement in regulating RAAS activity and Nlrp3-mediated innate immunity responses. Finally, we demonstrate that Ang (1–7) plays a protective function for the lung epithelial bronchoalveolar stem cells, enhancing their proliferation. This observation supports the potential role of Ang (1–7) in protecting lung damage during SARS-CoV-2 infection.

## Figures and Tables

**Figure 1 cells-11-02102-f001:**
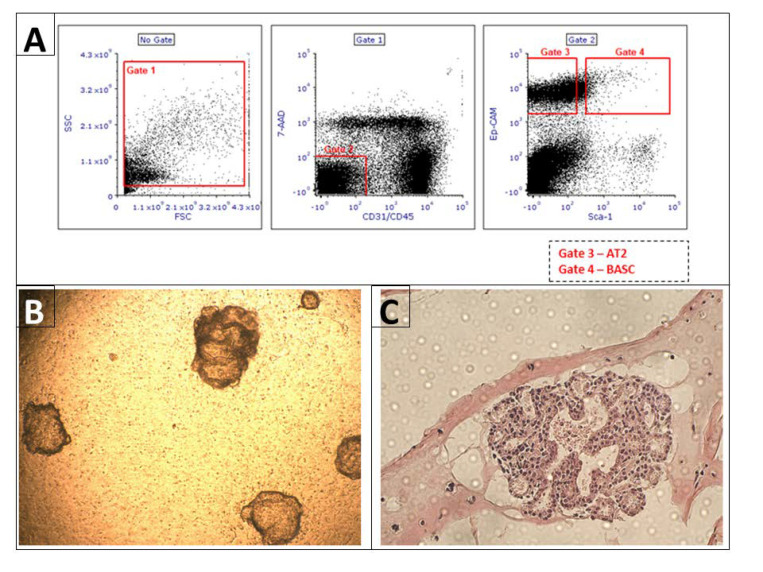
Isolation of alveolar type II cells and bronchioalveolar stem cells by FACS and 3D organoid culture started with FACS isolated cells carried out in a Matrigel. Panel (**A**): A representative plot showing the gating strategies used to isolate AT2 and BASC cells from mouse lungs. The gates were set to isolate in gate 3 AT2 (CD31 negative, CD45 negative, Ep-CAM positive, Sca-1 negative) and in gate 4 BASCs (CD31 negative, CD45 negative, Ep-CAM positive, Sca-1 positive). Panel (**B**): shows whole three-dimensional structure of organoid. These cultures are run for 21 days in the presence of pulmonary fibroblasts. Photograph of unstained organoid grown in insert was taken in brightfield with the use of light microscope at a 10× magnification. Panel (**C**): Organoid sections isolated at 21st day of culturing, paraffin embedded, 3 µm thick and H&E stained. Photograph was taken in brightfield at a 20× magnification.

**Figure 2 cells-11-02102-f002:**
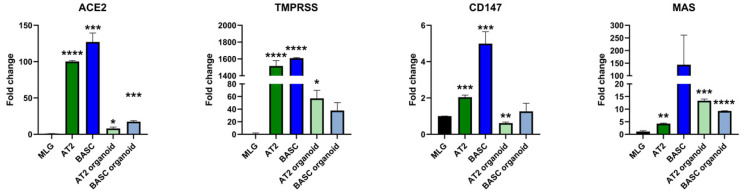
Relative mRNA expression of the genes involved in RAA system. Lung fibroblast cell line (MLG) was used as reference point for all examined genes expression. In order to analyze relative expression, comparative ΔCT method was employed. This method is a convenient way to calculate relative gene expression levels between different samples in that it directly uses the threshold cycles generated by the qRT-PCR system for calculation. * *p* < 0.05, ** *p* < 0.01, *** *p* < 0.001, **** *p* < 0.0001.

**Figure 3 cells-11-02102-f003:**
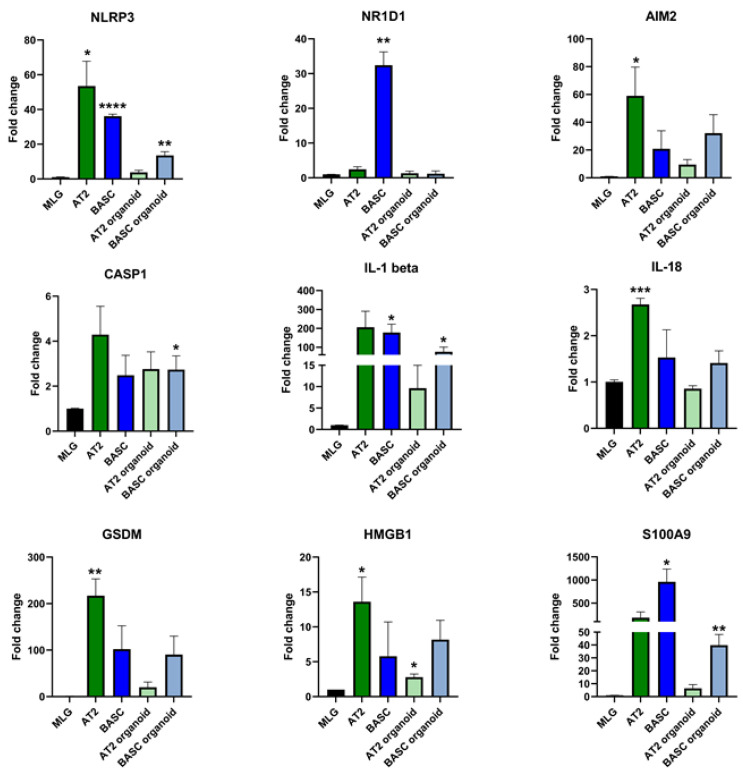
Relative mRNA expression of the genes involved in NLRP3 inflammasome. Lung fibroblast cell line (MLG) was used as reference point for all examined genes expression. In order to analyze relative expression, comparative ΔCT method was employed. This method is a convenient way to calculate relative gene expression levels between different samples in that it directly uses the threshold cycles generated by the qRT-PCR system for calculation. * *p* < 0.05, ** *p* < 0.01, *** *p* < 0.001, **** *p* < 0.0001.

**Figure 4 cells-11-02102-f004:**
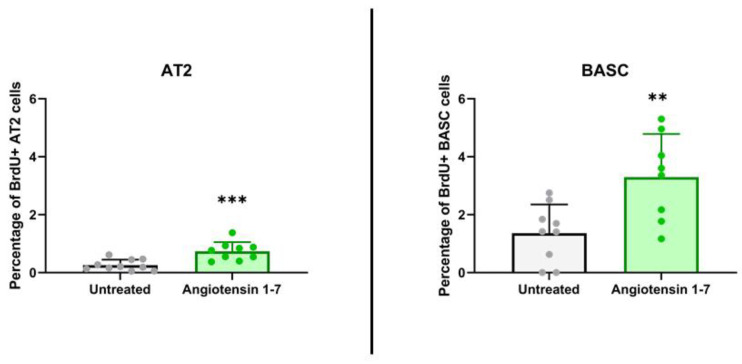
Murine alveolar type 2 cells and bronchoalveolar stem cells proliferate in vivo after stimulation by angiotensin 1–7. The graphs present incorporation of BrdU into AT2 cells and BASCs in response to 11-day administration of Angiotensin 1–7 in vivo. The percentage of AT2 cells or BASCs that showed proliferative activity (BrdU positive) were normalized to total number of the cells of these cell population. ** *p* < 0.01, *** *p* < 0.001 compared with control—not stimulated group.

**Figure 5 cells-11-02102-f005:**
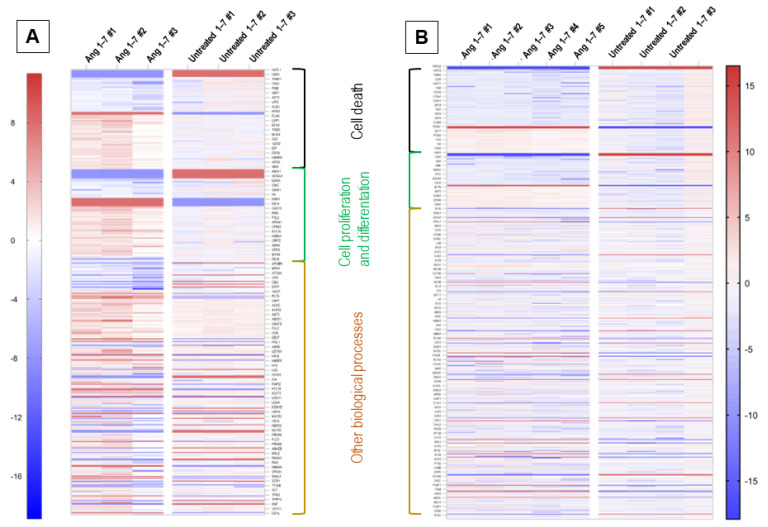
Heat map representing proteome expression changes analysis of identified proteins. Colors correspond to fold change of protein expression. Regions with certain biological processes are labeled in which represented genes are involved. Panel (**A**): BAL fluid protein profile expression changes after Ang (1–7) stimulation in comparison to BAL fluid isolated from untreated mice. Panel (**B**): Proteome expression observed in whole lung tissue lysate isolated from angiotensin (1–7) and untreated mice.

**Figure 6 cells-11-02102-f006:**
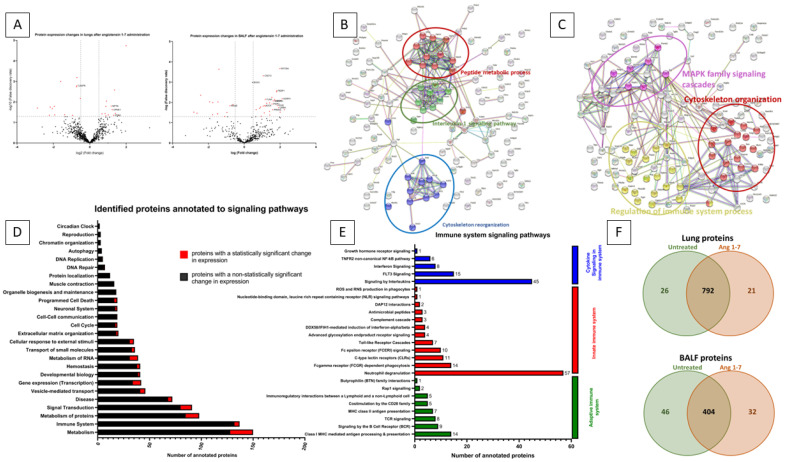
Proteome analysis of whole lung tissue lysate and bronchoalveolar lavage fluid. Panel (**A**): Volcano plot showing protein expression fold change values and their corresponding false discovery rate (−log10 *p*-value). Red points show statistically significant proteins with FDA > 1.3. Volcano plot on the left represents the whole lung tissue lysate proteome after angiotensin (1–7) stimulation normalized to untreated group. Volcano plot on the right represents the bronchoalveolar lavage (BAL) fluid proteome after angiotensin (1–7) stimulation normalized to untreated group. Panel (**B**): Graph of mutual interactions between proteins drawn with the use of SPRING.org. Only proteins with altered expression in whole lung tissue lysate after stimulation of mice with angiotensin (1–7) were included in the analysis. The main biological processes by which the identified proteins are functionally related are marked with separate colors. With red color are labeled proteins involved in peptide metabolic process, with green color are labeled proteins involved in Interleukin-1 signaling pathway, with blue color are labeled proteins involved in cytoskeleton reorganization. Panel (**C**): Graph of mutual interactions between proteins drawn with the use of SPRING.org. Only proteins with altered expression identified in BALF after stimulation of mice with angiotensin (1–7) were included in the analysis. The main biological processes by which the identified proteins are functionally related are marked with separate colors. With red color are labeled proteins involved in cytoskeleton organization, with purple color are labeled proteins involved in MAPK family signaling cascades, with yellow color are labeled proteins involved in regulation of immune system processes. Panel (**D**): Graph represents the number of identified proteins in whole lung tissue lysate which were annotated to signaling pathways based on Reactome.org. With black color are labeled proteins which were not show altered expression after Ang (1–7) stimulation compared to untreated group. With red color are labeled protein with statistically significant change in expression after Ang (1–7) stimulation, when compared to untreated group. Panel (**E**): Annotation to detailed signaling pathways of immune system divided into Cytokine signaling in immune system labeled with blue color, Innate immune system labeled with red color, Adaptive immune system labeled with green color. Panel (**F**): Venn diagram that shows total number of identified proteins in whole lung tissue lysate (upper diagram) and BALF (lower diagram) from untreated and Ang (1–7)-treated groups.

## Data Availability

Raw data is stored on local disks in the Laboratory of Regenerative Medicine and is available on request. In this matter, please contact the corresponding author.

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
