# Peer review of "Angiotensin 1–7 Stimulates Proliferation of Lung Bronchoalveolar Progenitors—Implications for SARS-CoV-2 Infection"

_cells, 2022, doi:10.3390/cells11132102_

Round 1

Reviewer 1 Report

The authors shows in the manuscript that lung epithelial stem cells, i.e. bronchoalveolar stem cells and alveolar type 2 cells, have entry receptors for SARS-Cov-2 virus, RAAS peptides and Nlrp3 inflammasome. They show the results proving that Ang 1-7 directly stimulates in vivo and in vitro proliferation of those two cell populations. They demonstrated that Ang 1-7 play a protective function for the lung epithelial bronchoalveolar stem cells, enhancing their proliferation. This observation supports the potential role of Ang 1-7 in protecting lung damage during SARS-Cov2 infection.

The strengths of this manuscript undoubtedly include well-chosen experimental methods and appropriate inference based on the obtained results. The authors also presented the results in an appropriate and clear manner in the form of figures. The presented results may have a significant impact on the development of the field of lung stem cell research. At the same time, as the authors themselves note, the topic requires further research.

However, in the reviewer’s opinion, the authors should explain why, out of 1,832 identified proteins, 836 proteins were analyzed. Moreover, authors should standardize the abbreviations they use, i.e. "Ang 1-7" is used in part of the manuscript and "Ang (1-7)" is used in another part of the manuscript.

Author Response

We would like to thank Reviewer 1 for its very positive evaluation of our research. As suggested by the Reviewer, we standardized "Ang (1-7)" and "angiotensin (1-7)" throughout the manuscript. In addition, we added in section 3.4 an explanation of the difference in the number of proteins identified versus the number of proteins that were subjected to the further analysis. The basis for the selection of proteins subjected to detailed proteomic analysis was their detection in experimental groups. Only proteins present in 51% of samples classified into one group were analyzed in detail further. Such an approach was taken to remove so-called individual differences between samples. Therefore, the proteomic profile was studied focusing on experimental group differences.

Reviewer 2 Report

Authors aimed to show that murine bronchioalveolar stem cells (BASC), alveolar type II cells (AT2), and 3D-derived organoids expression of mRNA encoding genes involved in virus entry into cells, components of RAA, and genes that comprise elements of Nlrp3 inflammasome pathway. Athors demonstrate that ACE-2 product Ang 1-7 stimulates proliferation of bronchioalveolar stem cells (BASC), alveolar type II cells (AT2), and 3D-derived organoids, which supports that it could be employed to ameliorate damage of lung alveolar stem/progenitor cells during SARS-Cov2 infection. Veyr complex research starting from mice,  through organoids to mRNA and protein expression on LC-MS. Lot's on collected data wiht clear results. Proper statistical analysis. It is a very good paper in my opinion.

Author Response

We would like to thank Reviewer 2 for appreciating the experimental design and the form of presenting the results obtained by our group.

Reviewer 3 Report

In the present manuscript the authors have presented a comprehensive study on the role of RAA pathway in maintaining homeostasis of lung bronchoalveolar cells.

The use of two different models  and unbiased approach to decipher the signaling crosstalk makes the study comprehensive and strengthen the hypothesis. 

section 2.1, The authors should mention the mean of animal body weight.

The author can move Table 1 in the supplementary.

The author should highlight the figure 6E in more detail in the results as well as in the discussion.

It will e good if the authors can add a schematic of RAA signaling pathway highlighted in the results section.

Author Response

Thank you for your suggestions on how we could improve our manuscript. As suggested by the Reviewer in section 2.1, we have added information about the average weight of the mice used in the experiment, that was 21 g (± 1 g).

Thank you for the suggestion to move the table to a supplement, which increased the clarity of the manuscript.

As suggested by the Reviewer, we have also described the results presented in Figure 6E in more detail.

To facilitate understanding of the RAS pathway, we have also added Figure 7, which shows a diagram of this signal pathway.